# Anisotropic Pinning-Effect of Inclusions in Mg-Based Low-Carbon Steel

**DOI:** 10.3390/ma11112241

**Published:** 2018-11-11

**Authors:** Chi-Kang Lin, Hsuan-Hao Lai, Yen-Hao Frank Su, Guan-Ru Lin, Weng-Sing Hwang, Jui-Chao Kuo

**Affiliations:** 1Department of Materials Science and Engineering, National Cheng Kung University, No. 1, University Road, Tainan 70101, Taiwan; cold19871025@gmail.com (C.-K.L.); lai57215308@gmail.com (H.-H.L.); wshwang@mail.ncku.edu.tw (W.-S.H.); 2Steelmaking Process Development Section, China Steel Corporation, No. 1, Zhonggang Road, Kaohsiung 81233, Taiwan; 150151@mail.csc.com.tw (Y.-H.F.S.); t120@mail.csc.com.tw (G.-R.L.)

**Keywords:** acicular ferrite, predict grain size, austenite grain size, pinning ability, Mg-based inclusion, low-carbon steel

## Abstract

In this study, the effect of austenite grain size on acicular ferrite (AF) nucleation in low-carbon steel containing 13 ppm Mg is determined. The average austenite grain size was calculated using OM Leica software. Results show that the predicted and experimental values of austenite grain size are extremely close, with a deviation of less than 20 µm. AF formation is difficult to induce by either excessively small and large austenite grain sizes; that is, an optimal austenite grain size is required to promote AF nucleation probability. The austenite grain size of 164 µm revealed the highest capacity to induce AF formation. The effects of the maximum distance of carbon diffusion and austenite grain size on the microstructure of Mg-containing low carbon steel are also discussed. Next, the pinning ability of different inclusion types in low-carbon steel containing 22 Mg is determined. The in situ observation shows that not every inclusion could inhibit austenite grain migration; the inclusion type influences pinning ability. The grain mobility of each inclusion was calculated using in situ micrographs of confocal scanning laser microscopy (CSLM) for micro-analysis. Results show that the austenite grain boundary can strongly be pinned by Mg-based inclusions. MnS inclusions are the least effective in pinning austenite grain boundary migration.

## 1. Introduction

Acicular ferrite (AF) is a ferrite phase that nucleates at inclusions of intragranular austenite grains [1]. Similar to bainite, AF possesses the same formation temperature range of 673–873 K and the same transformation process, including the incomplete-reaction phenomenon [2]. AF growth causes a shape deformation characterized by an invariant-plane strain with a large shear component. The AF microstructure is an important consideration in steel because the presence of this material can refine the microstructure and improve the toughness of steel [3,4]. These advantages are mainly due to the relatively high density of dislocations and fine-grained nature of the AF structure. The AF morphology includes needle-shaped crystallites and basket-weave structures [5].

According to some studies, AF formation depends on four factors: chemical composition and alloy element of materials [6,7], grain size of prior austenite [8], cooling rate during austenite transformation to ferrite [9,10], and inclusion type and inclusion size distribution [11,12,13]. Mortimer et al. reported that different chemical compositions and the presence of alloy elements, such as B, during ferrite nucleation may be involved in the change in grain boundary energy [14]. When Cr, Mo, or Ni contents increase, hardenability increases and AF structures are easy to form in low-carbon steel [15]. Cooling rate alters the microstructure of Δ*T* between the austenite (A3) transformation to the AF start nucleation temperature. Cooling can also result in thermal strain around inclusions and the matrix [16]. Yang et al. found that, with increasing Ti addition, cooling rates in the range of 0.5–2.5 °C/s yield the highest fraction of AF in medium-carbon steel [17]. Complex inclusions with MnS show more effective nucleation AF than simple inclusions with one phase [18]. Chai et al. found that adding Mg to Ti-Killed steels to form Ti–Mg–O inclusions can refine the inclusion size. Inducing AF nucleation is easier than inducing Ti–O inclusions [19].

Andrey et al. reported that heat treatment can affect AF nucleation. The impact of AF nucleation on inclusions and the volume fraction of the AF structure in steel increase with increasing surface area of inclusion particles (*A*_p_) and decreasing surface area of grain boundaries (*A*_GB_). The ratio of the surface areas of inclusion particles and grain boundaries (*A*_p_/*A*_GB_) is widely used to predict the probability of AF nucleation. Large surface areas of intraganular inclusions and low surface areas of grain boundaries are generally preferred [20]. In the present study, the relationships between austenite grain size and the probability of AF nucleation, as well as that between grain size and pinning ability of each inclusion type, are investigated.

## 2. Experiments

The specimen used in this study is low-carbon steel with minor Mg addition, the chemical composition of which is shown in Table 1. The specimens used for high-temperature confocal scanning laser microscopy (HT-CSLM) were machined into discs 8 mm in diameter and 3 mm in height and placed in a high-purity aluminum crucible. The sample preparation for austenite grain size used sandpaper to polish from number 240, 600, 1200, 2000 and 3000, and using diamond slurry to mirror polished from 3 and 1 μm. To determine the effect of austenite grain size AF nucleation, specimens of low-carbon steel containing 13 ppm Mg were used; these specimens were heated at 1473, 1573, or 1673 K for 30, 60, 180, or 300 s for austenitization. After heating, the specimens were cooled to room temperature at a cooling rate of 100 K/s. The average austenite grain size was calculated using OM Leica software based on ASTM E112. At least 300 grains for each condition were employed to measures average grain sizes and obtain reliable data. Finally, the microstructures of the specimens were observed after etching with 5% Nital and 95% alcohol solution by optical micrography (OMs).

To determine the pinning ability of different inclusion types, specimens of low-carbon steel containing 22 ppm Mg were used. Before heat treatment, SEM-EDS, EBSD, and ASPEX were utilized to determine the distribution of inclusion size, inclusion type, and positioning by Vickers. Then, specimens were heated at 1523 K for 600 s to calculate the velocity and driving force of each inclusion type during austenite grain boundary migration through HT-CSLM in situ observation.

## 3. Results and Discussion

### 3.1. Effect of Austenite Grain Size on the Formation of Induced Acicular Ferrite

Figure 1 shows the CSLM images of austenite grain microstructures in steel with 13 ppm Mg when heated at 1673, 1573, or 1473 K for 30, 60, 180, or 300 s. The austenite grain boundaries are clearly visible (black lines) in the CSLM images. The austenite grains were analyzed using OM Leica software (Grain Expert), and the average austenite grain sizes were calculated. Figure 2 shows the average austenite grain size obtained at different temperatures. The dashed line represents the fitting curves calculated using the grain growth equation $D=k_{0}\exp^{\left(\frac{-Q}{RT}\right)}t^{n}$, where *t* represents the time, *D* is the austenite grain size, *T* is the absolute temperature, *k*_0_ is a constant, *R* is the universal gas constant, and *Q* is the activation energy. Table 2 shows the parameters of the grain growth equation.

The austenite grain size was predicted from the grain growth equation. Heat treatment should be controlled to obtain the optimal austenite grain size. According to the calculation results, the austenite grain size was 57 µm when annealing was performed at 1673 K for 1 s. Thereafter, the time was calculated using the grain growth equation for 40 µm grain growth increments. Grain sizes of 88, 127, 169, 203, 239, and 276 µm were obtained by holding at 1673 K for 5, 20, 60, 120, 220, and 380 s, respectively.

In situ observations of austenite grains annealed at 1673 K for 1, 5, 20, 60, 120, 220, and 380 s are shown in Figure 3a–g. Austenite grain sizes of 51, 80, 112, 164, 192, 249, and 295 µm were obtained after holding for 1, 5, 20, 60, 120, 220, and 380 s, respectively. The typical OMs of the specimens during heat treatment at 1673 K for 1, 5, 20, 60, 120, 220, and 380 s are shown in Figure 4a–g. The microstructure of the sample grains consists of Widmanstatten ferrite, AF, bainite, and granular ferrite. Figure 5 shows the difference between the calculated and predicted values of austenite grain size. The error of austenite grain size is mainly below 15 µm, and the maximum error is only 19 µm, which is obtained from austenitization for 380 s. In general, increasing austenite grain sizes lead to reductions in the overall surface area of grain boundaries and increases in the volume fraction of the AF microstructure [1]. AF formation refines a coarse microstructure into a fine one and enhances the effect of grain refinement. Large austenite grains result in decreases in AF fraction. As seen in the OMs, additional inclusions induce AF with an appropriate austenite grain size. However, the volume fraction of AF is difficult to quantify precisely. The AF volume fractions of each sample under different austenitization conditions are difficult to distinguish and quantify from the micrographs. Thus, the probability of AF nucleation is used to quantify the ability of inclusions to induce AF at different austenitization holding times. The probability of pinning inclusions is defined by the following equation:(1)$$\mathrm{Probability}\;\mathrm{of}\;\mathrm{AF}\;\mathrm{nucleation}=\frac{N_{AF}}{N_{total}}\times 100\%$$
where *N_total_* and *N_AF_* are the total numbers of inclusions and inclusions with AF nucleation, respectively. After heat treatment, we used OM micrographs to find the microstructure of acicular ferrite around every inclusion, and separate the conclusions into two parts, one is inclusions without acicular ferrite formation, and the other is the inclusions with acicular ferrite formation. We used approximately 90–100 inclusions for statistical measurement of each sample for different austenitization holding times. The probabilities of AF nucleation from inclusions were 8%, 11%, 19%, 29%, 21%, 15%, and 5% for austenitization holding times of 1, 5, 20, 60, 120, 220, and 380 s, respectively.

Figure 6 shows the calculation results as a function of austenite grain size. The probabilities of AF nucleation for austenite grain sizes of 51, 80, 112, 164, 192, 249, and 295 µm were 8%, 11%, 19%, 29%, 21%, 15%, and 5%, respectively. This finding shows that the austenite grain size of 164 µm has the highest ability to induce AF nucleation, which means that the appropriate austenite grain size promotes AF formation. The austenite grain size changes the microstructure of steel [21]. Samples with small austenite grain sizes mainly consist of Widmanstatten ferrite, and small fractions of AF (Figure 4a–c). With increasing austenitizing holding time, the austenite grain size increases, and the main composition of the grains becomes Widmanstatten ferrite, and small fractions of AF.

### 3.2. Relationship between Diffusion Distance of Carbon and Austenite Grain Size

Previous research has verified that carbon concentration affects the nucleation potency of ferrite [1,22]. In general, low-carbon concentration areas allow the formation of considerably more ferrite than that formed in high-carbon concentration areas. The distribution of carbon over intragranular austenite grains may result in a low probability of ferrite formation inside the grain and easy formation of ferrite at the grain boundaries. By contrast, if carbon is distributed on the austenite grain boundary, ferrite easily forms inside the grain instead of at the grain boundaries. Widmanstatten ferrite formation affects the carbon distribution within the intragranular austenite, causing the carbon concentration to be non-uniform within intragranular austenite grains, which form before AF formation. However, carbon distribution cannot be determined by any analytical instrument. In the present study, we provide a theoretical calculation to explain the relationship between carbon distribution and probability of AF nucleation. The diffusion equation C is given as follows [23]:(2)$$C=\frac{M}{2\sqrt{\pi Dt}}\exp^{\left(-\frac{x^{2}}{4Dt}\right)}$$
where *D* is the diffusion coefficient, *t* is the time, *x* is the diffusion distance, and *M* is a known quantity. *D* is influenced by temperature. The diffusion coefficient equation is expressed as follows [23]:(3)$$D=D_{0}e^{\left(-\frac{Q}{RT}\right)}$$
where *D*_0_ is the maximal diffusion coefficient, *Q* is the activation energy, *T* is the absolute temperature, and *R* is the universal gas constant. Considering that Widmanstatten ferrite affects the carbon distribution in intragranular austenite grains is difficult. Thus, we simply consider the carbon distribution in intragranular austenite grains and calculate the value that can attain the maximum diffusion distance. When the austenite transforms to ferrite, the driving force causes carbon diffusion from the inner region of the austenite grains to the austenite grain boundary. In our hypothesis, the carbon concentration of intragranular austenite grains is 0 wt.%, at which point carbon diffuses to the austenite grain boundary. Therefore, the value of *M* is 0.13 wt.%.

Figure 5 also shows the relationship between the maximum carbon diffusion distance and annealing time. The calculated result shows that, despite increases in time, the maximum distance of carbon diffusion remains the same. Lin et al. [24] observed the AF formation time at high temperatures. The time from the start of AF formation to completion was 10 s. This finding means the carbon diffusion time is limited by the AF formation time. Based on Equation (2), the calculated maximum distance of carbon diffusion is 71 µm. Over this distance, a carbon-free zone with a diameter of 142 µm will be formed.

The grain is separated into carbon-rich and carbon-free areas via the boundary line of maximum distance of carbon (Figure 7a). Ferrite formation is easily induced in the carbon-free area. Figure 7b shows that when the grain size is smaller than the maximum distance of carbon diffusion, carbon diffuses to other grains. Thus, adjacent grains retain their carbon-rich condition. As such, AF formation is difficult to induce in extremely small austenite grains. If the austenite grain size is similar to the maximum carbon diffusion distance, intragranular austenite grains become carbon-free and can easily induce AF formation.

When the austenite grain size is larger than the maximum distance of carbon diffusion, the carbon concentration of intragranular austenite grains is higher than that at the austenite grain boundary. Ferrite consequently exhibits increased nucleation potency at the grain boundary rather than inside austenite grains. Thus, an excessively large austenite grain results in a low probability of AF nucleation because of the carbon-rich area distribution in grains. According to the discussion above, AF nucleation can be promoted by the appropriate austenite grain size. An excessively small or large austenite grain size may not be the optimal grain size for AF nucleation. The calculation results of carbon diffusion distance and its effect on AF formation are shown in Figure 6.

If the distance from the center of the grain to grain boundary is similar to the maximum carbon diffusion distance, AF is easily induced owing to the carbon-free grains. The calculated maximum distance of carbon diffusion is 71 µm, and the corresponding austenite grain size, considering a spherical austenite grain, is 142 µm. In Figure 6, the probability of AF nucleation increases with increasing austenite grain size, but an excessively large austenite grain reduces the probability of AF formation. The optimized austenite grain size with the highest probability of AF nucleation is 164 µm, which is relatively similar to the calculation results of grain size with the corresponding maximum carbon diffusion distance.

### 3.3. Inclusion Behavior

Zener et al. reported that grain growth can be retarded by inclusions [25]. Our previous discussion revealed that AF formation is affected by the grain size of austenite. Thus, understanding the anisotropic pinning effect of inclusions on austenite grain growth is important. Figure 8 demonstrates the interaction between austenite grains and inclusions in low-carbon steel containing 22 ppm Mg as an example. Here one austenite grain boundary passes through an inclusion in Figure 8a–c, whereas another austenite grain boundary is retarded by an inclusion in Figure 8d–f. The results obtained suggest that austenite grain boundaries can pass through an inclusion or be pinned by it. In general, inclusion could inhibit the austenite grain boundaries growth. In this study, we observed that it’s not every inclusion could inhibit austenite grain growth from in-situ observation of CSLM. This variation inspires us to investigate the relationship between the anisotropic pinning effect and inclusions.

EDS and EBSD were utilized to determine inclusions in low-carbon steel samples containing 22 ppm Mg. The EDS spectra and Kikuchi patterns of the inclusions are summarized in Figure 9; these inclusions include MgO, MgO·Al_2_O_3_, MnS, MgO–MnS, and MgO·Al_2_O_3_–MnS. Here MnS is able to precipitate on MgO or MgO·Al_2_O_3_ to form complex inclusions, such as MgO–MnS or MgAl_2_O_4_-MnS [26]. ASPEX measurements were used to characterize the number and type of inclusions, and Figure 10a–e show the histograms and cumulative frequency distributions of these inclusions. Most of the MgO-MnS inclusions range in size from 3 µm to 4 µm and from 5 µm to 10 µm in Figure 10c. By comparison, MnS inclusions range in size from 1 µm to 3 μm in Figure 10e.

After identifying inclusions, the mobility of grain boundaries was determined using in situ micrographs obtained by HT-CSLM. Here the mobility m of the grain boundaries is proportional to the net driving force, which is equal to the driving force of grain growth (*F_P_*) subtracted from the Zener pinning force (*F_R_*) and given as follows:(4)$$V=\left(F_{P}-F_{R}\right)m=\Delta F\cdot m$$
where *V* is the velocity of the austenite boundary. According to Burke et al. report, the driving force for austenite grains is expressed as follows [27]:(5)$$F_{p}=\frac{2\sigma_{b}}{R}$$
and the retarding force is given as
(6)$$F_{R}=\frac{3f_{V}\sigma_{b}}{2r}$$
where *R* is the curvature of the grain boundary, $\sigma_{b}$ is the austenite grain boundary energy, *f_V_* is the volume fraction of the inclusion, and r is the inclusion radius. Here, the radius *r* is equal to half of the average diameter, and *f_A_* is obtained by dividing the number of inclusions *N* by the measured area *A*. Then, the equation *f_V_* = *f_A_*/2*N* is employed.

To determine the driving force for austenite grain, the average curvature of the grain is calculated by adopting the method proposed by Aleksandra et al. report [28]. Considering the boundaries ac and cb in Figure 11, we combined two points, a and b, and obtained a triangle acb. Then, the vertical lines of ac, cb, and ab are OC’, OD’, and OB’, respectively. Subsequently, point O was determined as the point of intersection of lines OC’, OD’, and OB’. Thus, the vertical distance from the point of intersection O to ac and cb is the curvature R of the grain boundaries ac and cb. The inclusion radius r was directly observed from the in situ image of HT-CSLM. Here the austenite grain boundary energy $\sigma_{b}$ is 1.159 J/m^2^, as reported by Vynokur [29]. With these data, we were able to calculate *F_R_*, *F_P_*, and *V* according to Equations (5), (6), and (4), respectively.

Based on the results of HT-CSLM images, inclusions were found at the interior of grains, at the boundaries, and at the triple points. In theory, we should have identified inclusions by EDS first, and these inclusions should lie at the same time at the boundaries or at the triple points after annealing because the grain structure appears after annealing due to the thermal grooving effect. However, we successfully observed 17 inclusions at the boundaries or triple points and summarized our observations as four cases in Table 3 for MgO, MnS, MgO–MnS, and MgO·Al_2_O_3_–MnS. Figure 12, Figure 13, Figure 14, Figure 15 and Figure 16 show examples of in situ HT-CSLM micrographs of inclusions of MgO, MnS, MgO-MnS, and MgO·Al_2_O_3_-MnS. The grain boundaries are retarded by inclusions in the first case, which we call the pinning grain boundary The triple junctions of grain boundaries are pinned by inclusions in the second case, which we call the pinning triple point. The pinned grain boundaries pass through inclusions later in the third case, which we call the passing-through grain boundary, and the pinned triple junction of grain boundaries passes through inclusions in the fourth case, which we call the passing-through triple junction.

Three pinning boundaries, one pinning triple junction, and one passing-through boundary are observed for MgO in Table 3. The grain boundary is retarded by MgO inclusion in Figure 12a–c, and the velocity of the austenite boundary is 0 μm/min in Figure 12d. Figure 12e–g show the pinned triple junction with a boundary velocity of 0 µm/min in Figure 12h. Figure 12i–k illustrate boundary migration with inclusions. First, the austenite grain boundary without inclusions migrates freely, and the boundary velocity is 1.085 µm/min at annealing times less than 367 s. Subsequently, the boundary is pinned by inclusions, and the boundary velocity drops to 0 µm/min between annealing times of 367 and 604 s. Then, when the annealing time exceeds 604 s, the inclusions are unable to retard the migration of austenite grains. Grain boundary migration occurs through the inclusions, and the boundary velocity is 0.5838 µm/min, as shown in Figure 12l.

Four passing-through grain boundaries and one pinning triple junction are observed for MnS in Table 3. Figure 13a–c show the passing-through grain boundary. The boundary without inclusions can move freely when the annealing time is less than 484 s [Figure 13a]. The austenite grain boundary is pinned by inclusions between annealing times of 484 and 690 s [Figure 13b]. After annealing for 690 s, the boundary passed through the inclusions and continues to grow [Figure 13c]. Figure 13d shows the evolution of the boundary velocity from 1.58 µm/min to 0 μm/min and then to 1.06 μm/min. Figure 13e–g show the pinning triple junction. First, the triple junction of the grains has a boundary velocity of 6.71 µm/min at annealing times less than 348 s and 12.597 μm/min between annealing times of 348 and 401 s in Figure 13e,f, respectively. After annealing for 484 s, the boundary is pinned, as shown in Figure 13g.

One pinning grain boundary is observed for MgO·Al_2_O_3_ in Table 3. Figure 14a–c show that the boundary was pinned by inclusion, even when the annealing time increased to 440 s, and the boundary velocity was 0 µm/min, as shown in Figure 14d.

One pinning grain boundary, two pinning triple junctions, and one passing-through grain boundary are observed for MgO–MnS in Table 3. Figure 15a shows the pinning grain boundary. At annealing times of less than 563 s, the inclusions retard the boundary, as shown in Figure 15b,c, and the boundary velocity is 0 µm/min in Figure 15d. Figure 15e shows the triple junction pinned by inclusions. When the annealing time is increased to 406 s, the triple point of the grain boundaries remains pinned by inclusions, as shown in Figure 15f,g, and the boundary velocity is 0 µm/min, as shown in Figure 15h. Figure 15i–k illustrate the passing-through grain boundary. First, the boundary without inclusions migrates freely, and the boundary velocity is 2.29 µm/min at annealing times of less than 410 s. Next, the austenite grain boundary is retarded by inclusions at annealing times between 540 to 573 s. After annealing for 573 s, the boundary moves with a boundary velocity of 1.93 µm/min and passes through the inclusion, as shown in Figure 15l.

One pinning grain boundary, one pinning triple point, and one passing-through grain boundary are observed for MgO·Al_2_O_3_–MnS (Table 3). The grain boundary is pinned by MgO·Al_2_O_3_–MnS inclusions, as shown in Figure 16a–c. Even when the annealing time increases to 592 s, inclusions continue to pin the boundary, and the boundary velocity is 0 µm/min, as shown in Figure 16d. Figure 16e shows a triple junction of grain boundaries retarded by inclusions. When the annealing time is increased to 641 s, the triple junction of the grain boundaries remains retarded by these inclusions, as shown in Figure 16f–g, and the boundary velocity is 0 µm/min in Figure 16h. Figure 16i–k illustrate the process of the passing-through grain boundary. The boundary without inclusions could migrate freely with a boundary velocity of 4.53 μm/min annealing times of less than 231 s, as shown in Figure 16i. At annealing times between 231 and 410 s, the boundary has a velocity of 8.06 μm/min and passes through the inclusion in Figure 16j. When the annealing time exceeds 464 s, the boundary does not move (boundary velocity, 0 μm/min), as shown in Figure 16l.

According to the results described above, the mobility of austenite grain boundaries can be calculated using Equation (4), together with the parameters measured from the HT-CSLM micrographs. However, in this work, we simply calculated two cases of the passing-through boundary and the passing-through triple junction. Figure 17 shows the boundary velocity in terms of the net force ΔF for MgO, MnS, MgO-MnS, and MgO·Al_2_O_3_–MnS. The mobility of austenite grain boundaries was calculated using Equation (4) as the slope in Figure 17. From Equation (4), when the grain boundary and triple point are pinned by an inclusion and the ∆*V* of grain growth is 0 µm/min, the grain mobility is 0. Table 4 lists all grain mobilities, boundary velocities, and ∆*F* of the inclusions. The grain mobilities of MgO, MgO·Al_2_O_3_-MnS, and MgO–MnS were 0.05, 0.20, and 0.31, respectively. The average grain mobility of the MnS inclusion is 0.41 because this inclusion has four passing-through grain boundaries. Based on these results, Mg-based inclusions have lower grain mobility than MnS inclusions. From Equation (4), the degree of pinning effect gets higher with smaller value of grain mobility. Among the Mg-based inclusions, MgO inclusions showed the lowest grain mobility; in other words, MgO inclusions effectively retard austenite grain growth while MnS inclusions do not. It means that the MgO inclusions have the best pinning effect than other inclusions.

## 4. Conclusions

In this work, austenite grain size was predicted from the grain growth equation, and the error of austenite grain size was determined to be less than 20 µm. The heat treatment condition at 1673 K for 60 s has the highest probability of AF nucleation. The probability of AF nucleation increased with austenite grain size, but an overly large austenite grain decreased the probability of AF nucleation because of the carbon concentration effect. The maximum distance of carbon diffusion affected the formation probability of AF in steel and denoted the relationship between the optimized austenite grain size for AF formation and the final microstructure of the steel. An austenite grain size of 164 µm was optimal for inducing the formation of AF in low-carbon steel containing 13 pm Mg. Mg-based inclusions were more effective in retarding austenite grain growth than MnS inclusion due to the low mobility of grain growth of the former. MgO inclusions also showed the best pinning efficiency.

## Figures and Tables

**Figure 1 materials-11-02241-f001:**
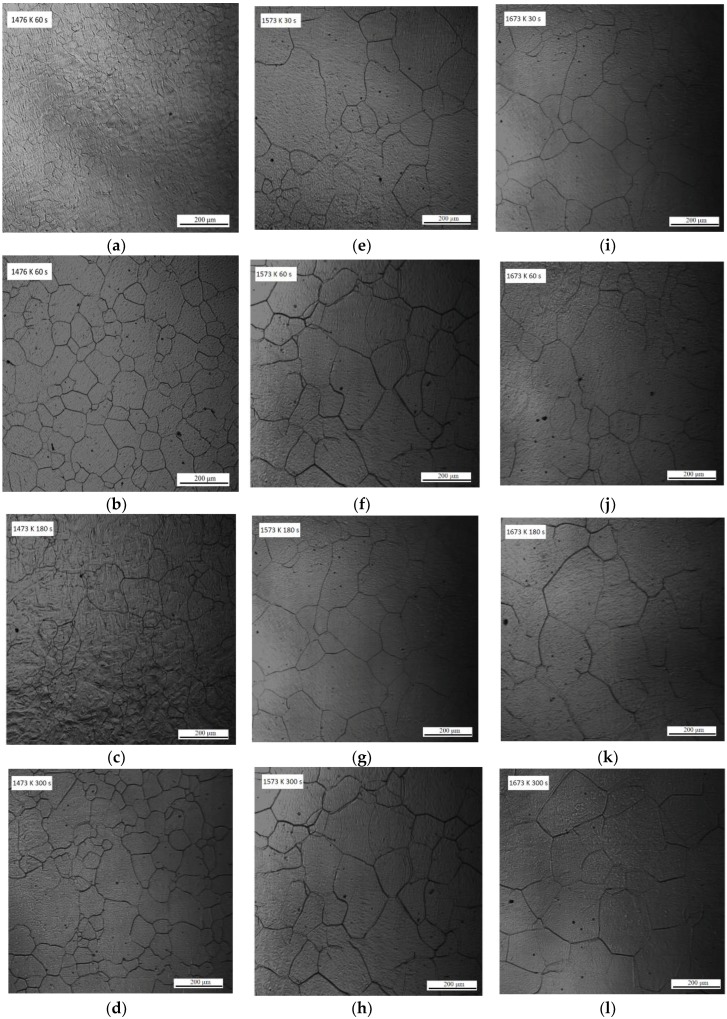
HT-CSLM images of austenite grains in low-carbon steel with 13 ppm Mg annealed for (**a**,**e**,**i**) 30 s, (**b**,**f**,**j**) 60 s, (**c**,**g**,**k**) 180 s, and (**d**,**h**,**l**) 300 s. (**a**–**d**) at of 1473 K, (**e**–**h**) at 1573 K, and (**i**–**l**) at 1673 K.

**Figure 2 materials-11-02241-f002:**
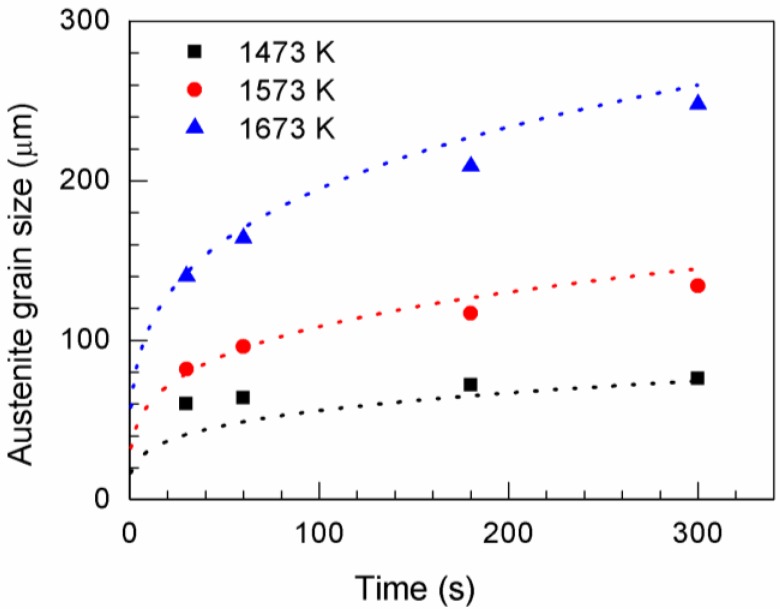
Average grain size of austenite as a function of annealing time in low-carbon steel with 13 ppm Mg at 1473, 1573, and 1673 K. Symbols indicate the experimental data, and the fitting curves are indicated by dashed lines.

**Figure 3 materials-11-02241-f003:**
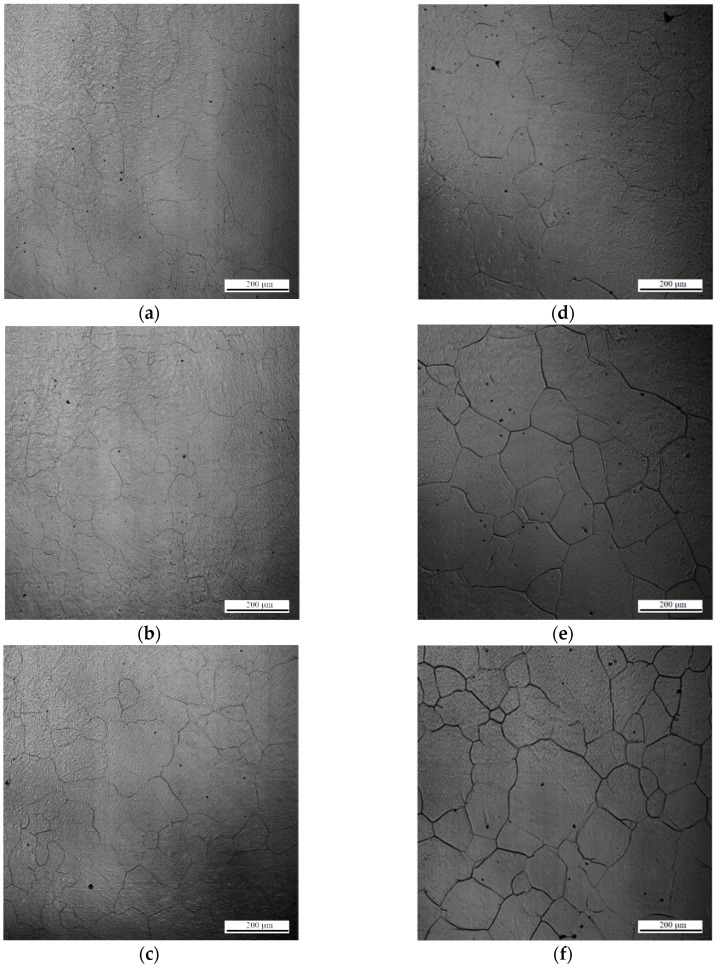
HT-CSLM images of austenite grain in low-carbon steel with 13 ppm Mg at 1673 K for (**a**) 1 s, (**b**) 5 s, (**c**) 20 s, (**d**) 60 s, (**e**) 120 s, (**f**) 220 s and (**g**) 380 s.

**Figure 4 materials-11-02241-f004:**
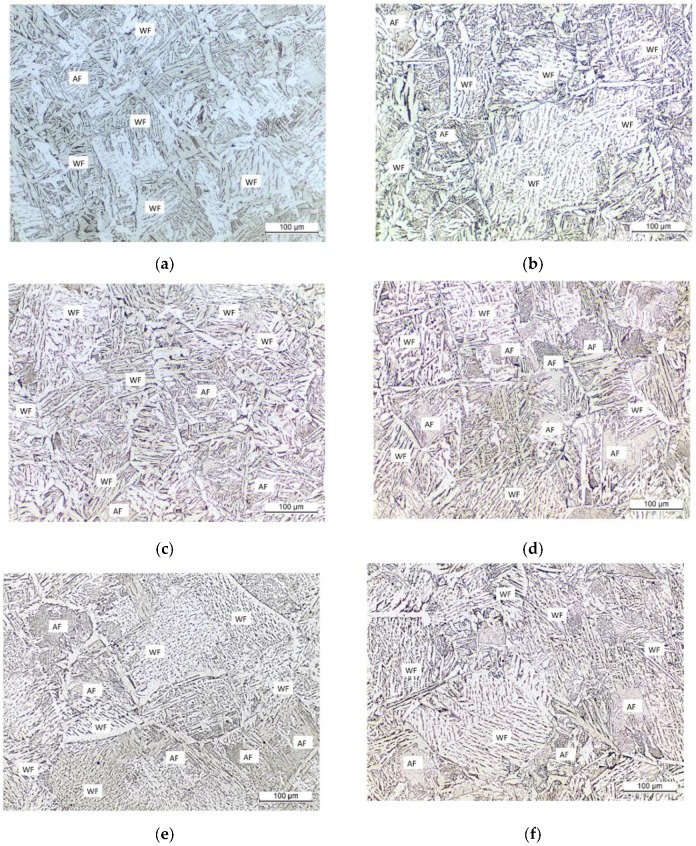
Optical micrographs of low-carbon steel with 13 ppm Mg heated at 1673 K for (**a**) 1 s, (**b**) 5 s, (**c**) 20 s, (**d**) 60 s, (**e**) 120 s, (**f**) 220 s and (**g**) 380 s. (WF is Widmanstatten ferrite, and AF is acicular ferrite).

**Figure 5 materials-11-02241-f005:**
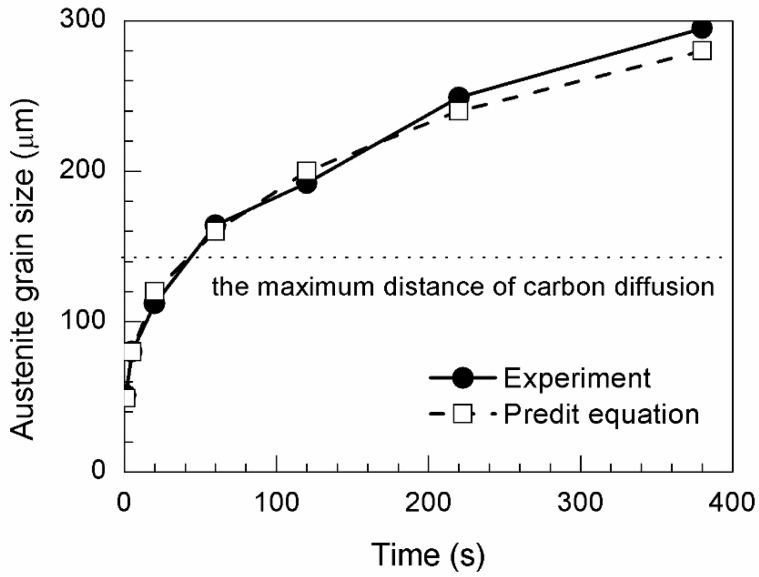
Comparison of average grain size of austenite between experimental data and the fitting function in terms of annealing time in low-carbon steel with 13 ppm Mg at 1673 K for different holding time.

**Figure 6 materials-11-02241-f006:**
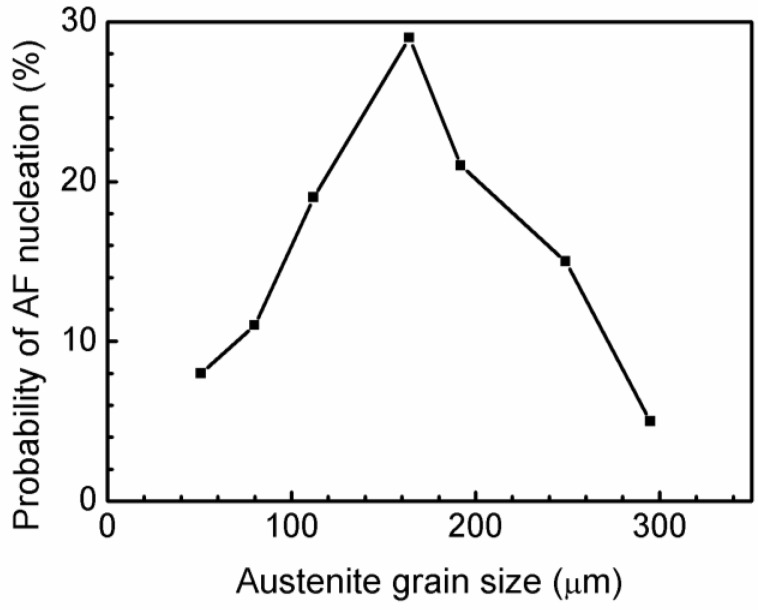
Probability of AF nucleation in terms of austenite grain size.

**Figure 7 materials-11-02241-f007:**
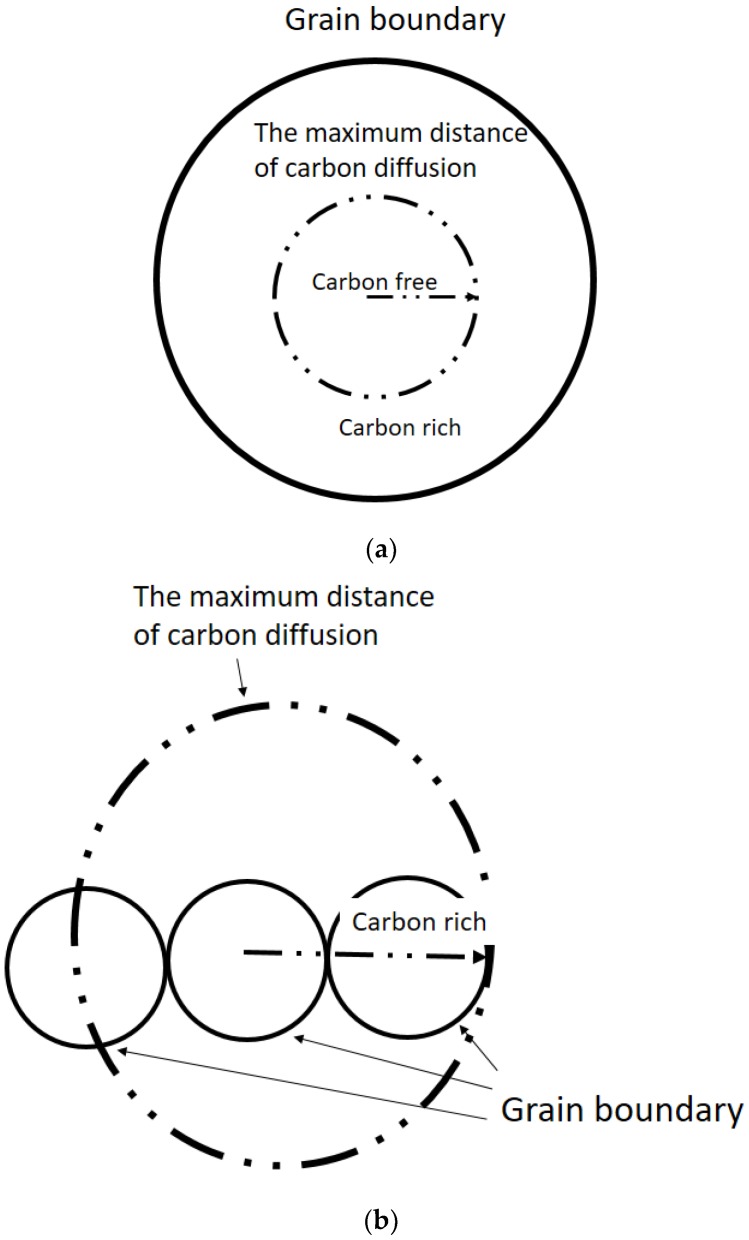
Schematic illustration of the maximum distance of carbon diffusion for (**a**) large grain size and (**b**) small grain size.

**Figure 8 materials-11-02241-f008:**
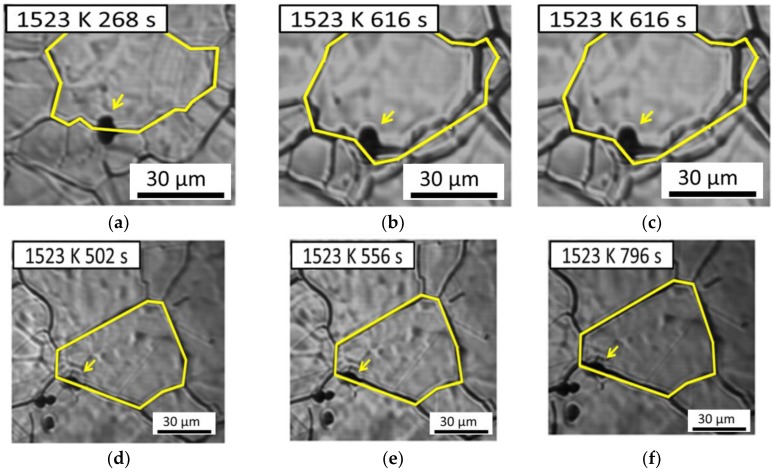
In situ observation of austenite grain boundaries pinned (**a**–**c**) without and (**d**–**f**) with an inclusion of low-carbon steel containing 22 ppm Mg. Arrows indicate the inclusion and the yellow line indicates austenite grain boundaries.

**Figure 9 materials-11-02241-f009:**
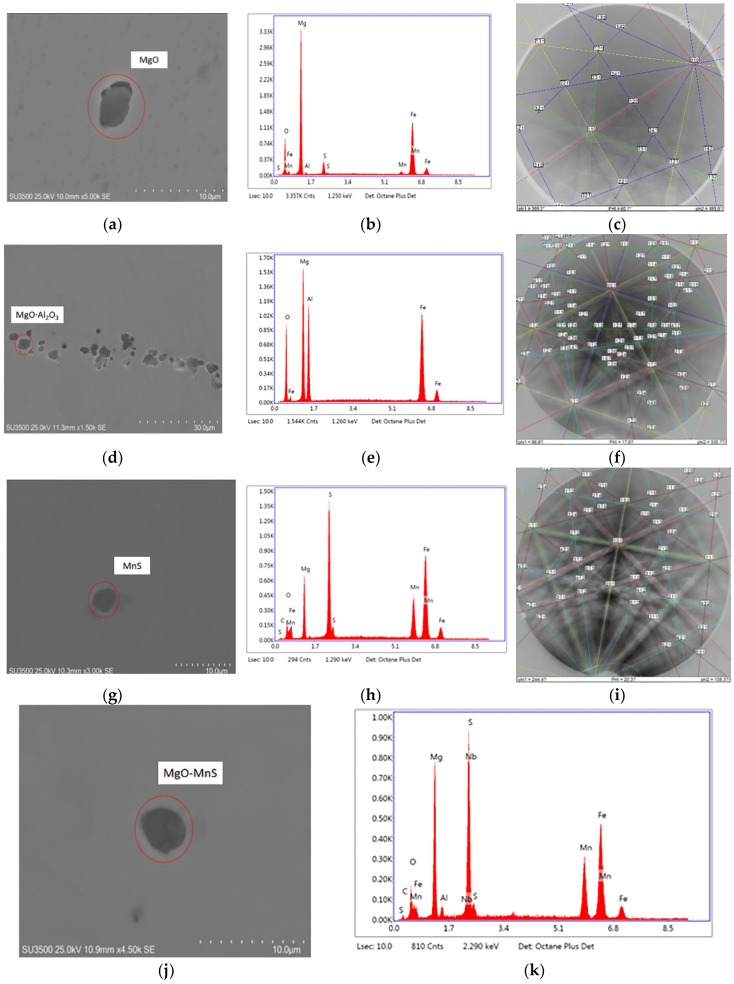
Inclusion analysis of low-carbon steel containing 22 ppm Mg. (**a**,**d**,**g**,**j**,**l**) SEM micrographs, (**b**,**e**,**h**,**k**,**m**) EDS spectra, and (**c**,**f**,**i**) Kikuchi patterns. (**a**–**c**) MgO inclusions, (**d**–**f**) MgO·Al_2_O_3_ inclusions, (**g**–**i**) MnS inclusions, (**j**–**k**) MgO–MnS inclusions, and (**l**–**m**) MgO·Al_2_O_3_–MnS inclusions.

**Figure 10 materials-11-02241-f010:**
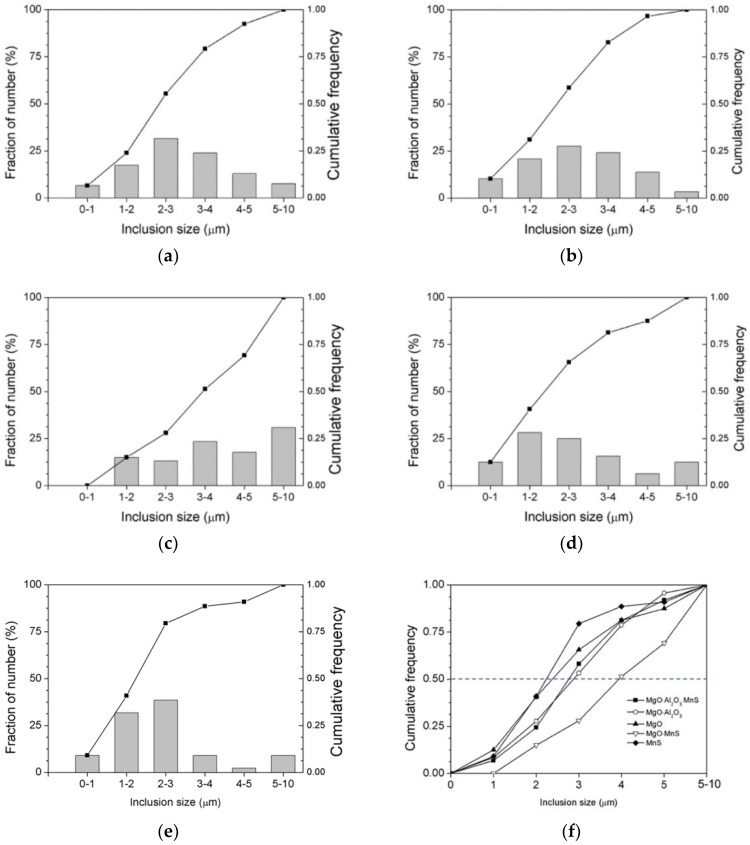
Histograms and cumulative frequency distributions of inclusions of (**a**) MgO·Al_2_O_3_–MnS, (**b**) MgO· Al_2_O_3_, and (**c**) MgO–MnS, (**d**) MgO, (**e**) MnS, and (**f**) all inclusions in low carbon steel containing 22 ppm Mg.

**Figure 11 materials-11-02241-f011:**
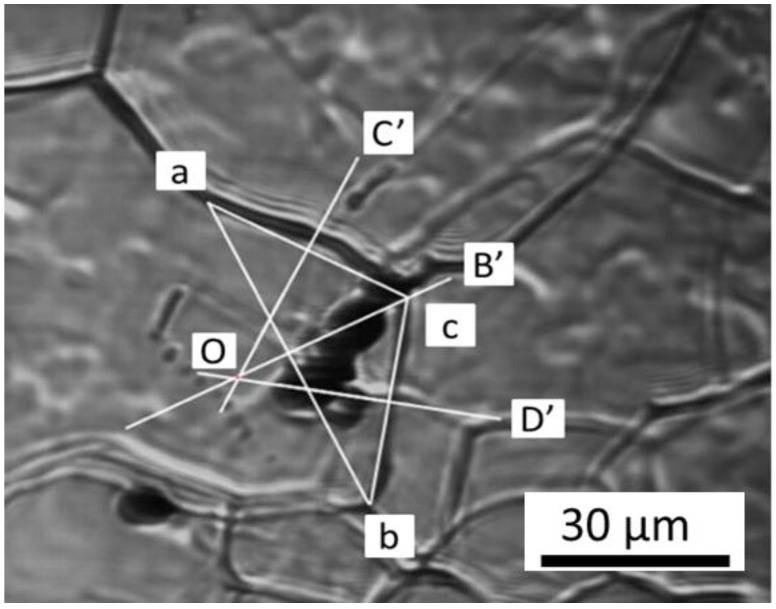
Determination of the central curvature of one grain.

**Figure 12 materials-11-02241-f012:**
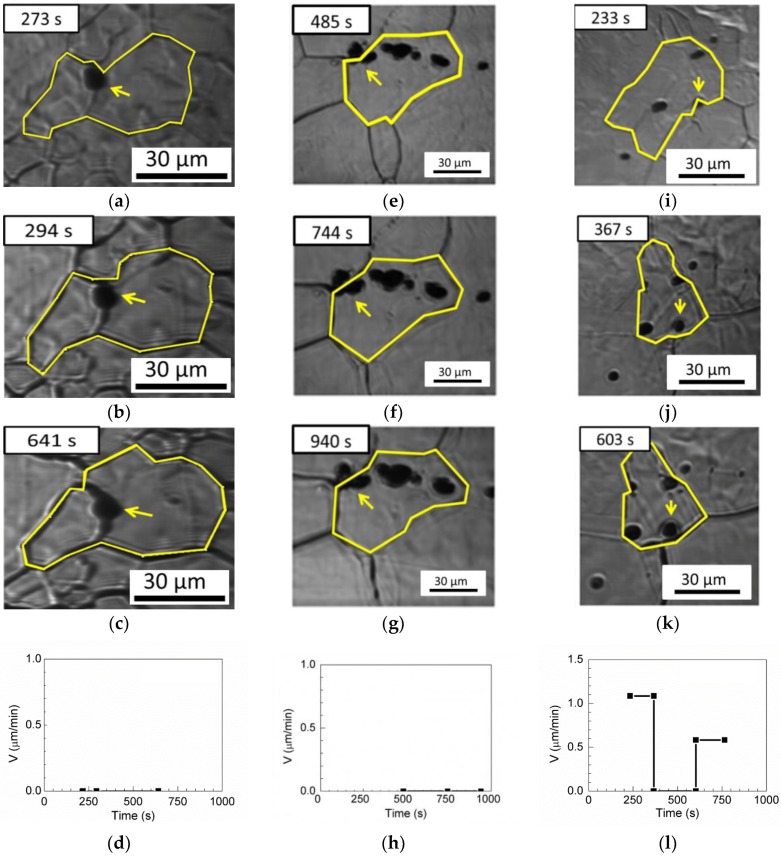
In situ HT-CSLM images of grain boundaries pinned by MgO inclusions at (**a**) 273 s, (**b**) 294 s, and (**c**) 641 s. In situ CSLM images of triple-point pinning by MgO inclusions at (**e**) 485 s, (**f**) 744 s, and (**g**) 940 s. In situ CSLM images of grain boundary transgranular by MgO inclusions at (**i**) 233 s, (**j**) 367 s, and (**k**) 603 s. Boundary velocities and driving forces as functions of time in (**d**) the pinning boundary, (**h**) the pinning triple junction, and (**l**) the passing-through boundary. Arrows indicate inclusions Arrows indicate inclusions, yellow line indicate austenite grain boundaries.

**Figure 13 materials-11-02241-f013:**
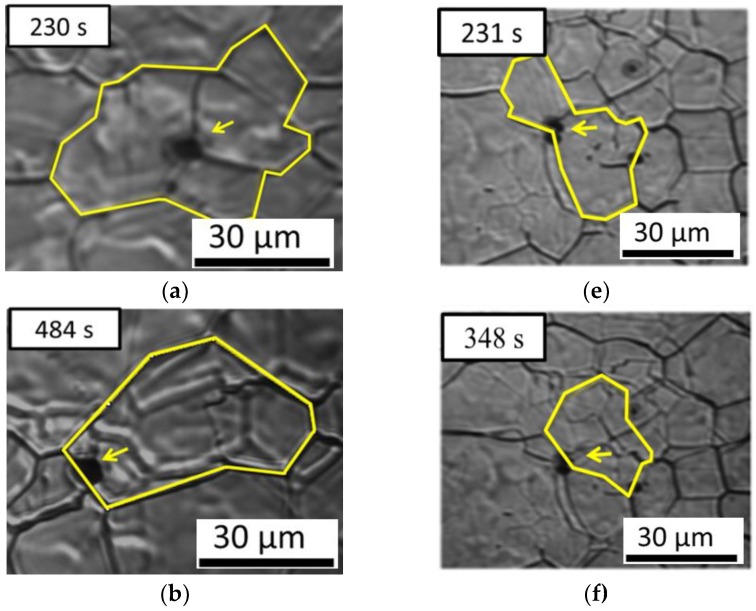
In situ CSLM images of grain boundary transgranular by MnS inclusions at (**a**) 230 s, (**b**) 484 s, and (**c**) 690 s. In situ CSLM images of triple-point transgranular by MnS inclusion at (**e**) 231 s, (**f**) 348 s, and (**g**) 401 s. Boundary velocities and driving forces as functions of time in (**d**) the passing-through triple junction and (**h**) the passing-through boundary. Arrows indicate inclusions, yellow line indicate austenite grain boundaries.

**Figure 14 materials-11-02241-f014:**
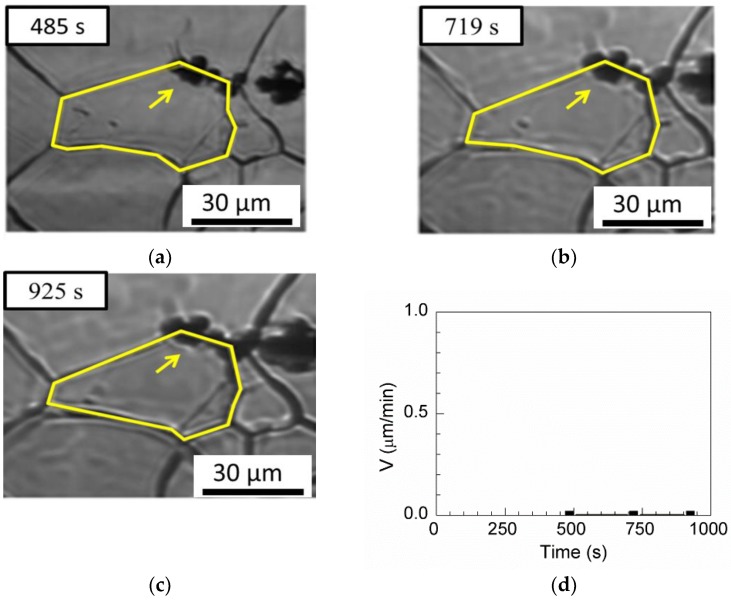
In situ CSLM images of grain boundary pinning by MgO∙Al_2_O_3_ inclusions at (**a**) 485 s, (**b**) 719 s, and (**c**) 925 s. Boundary velocities and driving forces as functions of time in (**d**) the pinning boundary. Arrows indicate inclusions, yellow line indicate austenite grain boundaries.

**Figure 15 materials-11-02241-f015:**
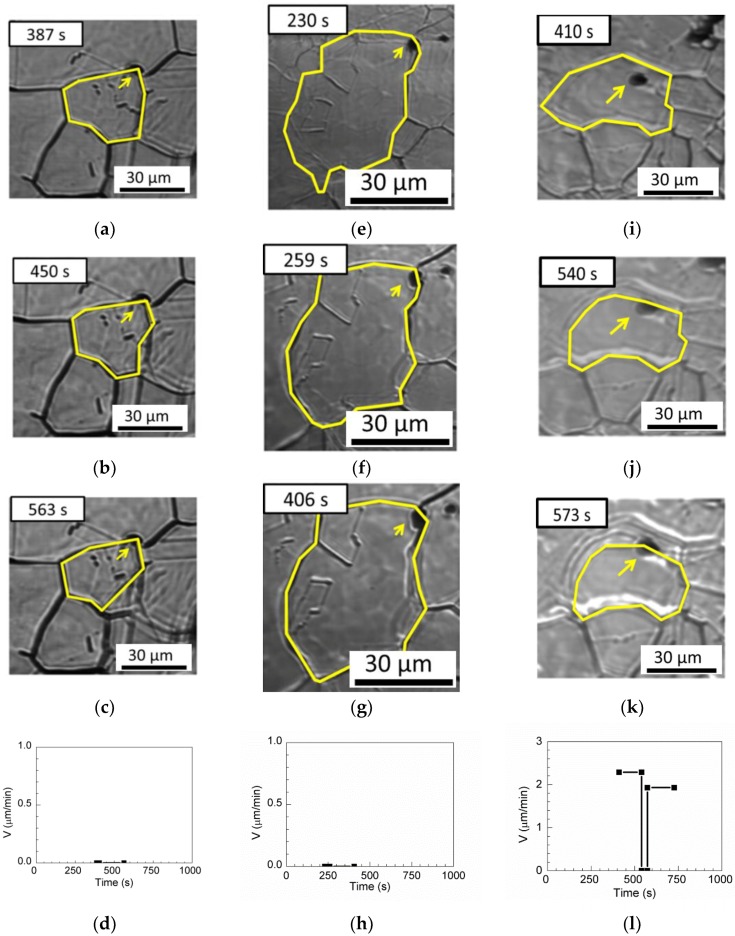
In situ CSLM images of grain boundary pinning by MgO-MnS inclusion at (**a**) 387 s, (**b**) 450 s, and (**c**) 563 s. In-situ CSLM images of triple-point pinning by MgO-MnS inclusion at (**e**) 230 s, (**f**) 259 s, and (**g**) 406 s. In situ CSLM images of grain boundary transgranular by MgO-MnS inclusion at (**i**) 410 s, (**j**) 540 s, and (**k**) 573 s. Boundary velocities and driving forces as functions of time in (**d**) the pinning boundary, (**h**) the pinning triple junction, and (**l**) the passing-through boundary. Arrows indicate inclusions, yellow line indicate austenite grain boundaries.

**Figure 16 materials-11-02241-f016:**
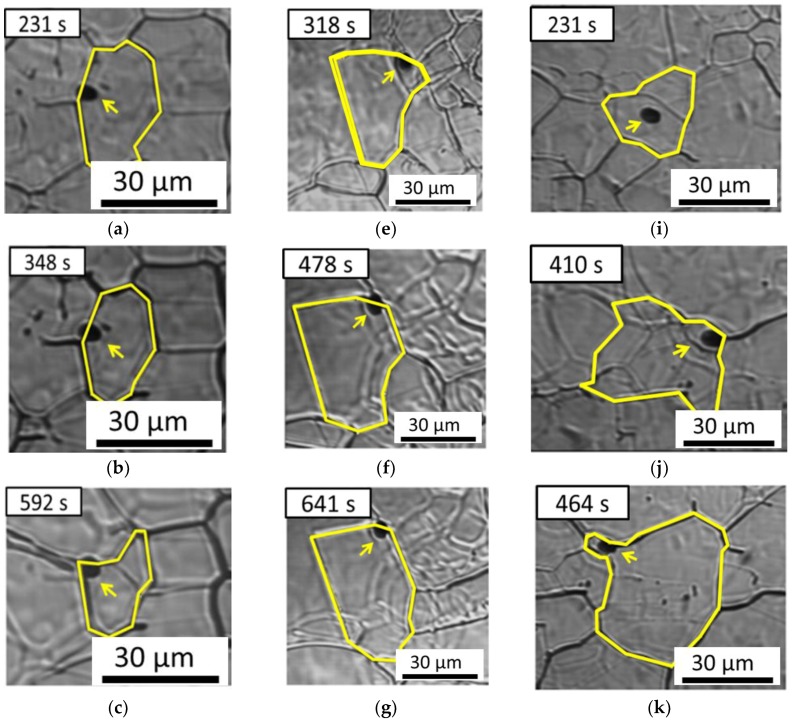
In situ HT-CSLM images of grain boundary pinning by MgO·Al_2_O_3_–MnS inclusions at (**a**) 231 s, (**b**) 348 s, and (**c**) 592 s. In situ CSLM images of triple-point pinning by MgO∙Al_2_O_3_–MnS inclusions at (**e**) 318 s, (**f**) 478 s, and (**g**) 641 s. In situ HT-CSLM images of grain boundary transgranular by MgO·Al_2_O_3_–MnS inclusion at (**i**) 231 s, (**j**) 410 s, and (**k**) 464 s. Boundary velocities and driving forces as functions of time in (**d**) the pinning boundary pinning, (**h**) the pinning triple point, and (**l**) the passing-through boundary. Arrows indicate inclusions, Arrows indicate inclusions, yellow line indicate austenite grain boundaries.

**Figure 17 materials-11-02241-f017:**
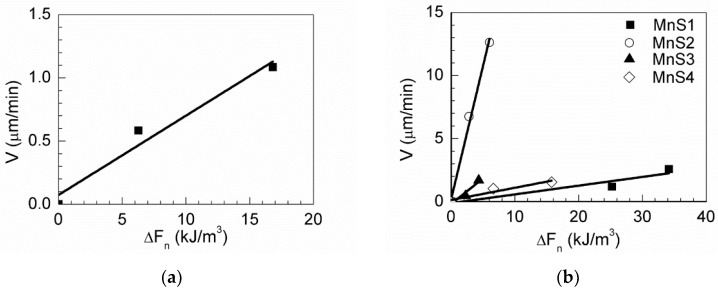
Boundary velocity as a function of ∆*F* (FP-FR) for (**a**) MgO, (**b**) MnS, (**c**) MgO–MnS, and (**d**) MgO·Al_2_O_3_–MnS. FP is the driving force of grain growth, and FR is the retarding force on boundaries.

**Table 1 materials-11-02241-t001:** Chemical composition of the studied low carbon steel (wt.%).

C	Si	Mn	P	S	N	O	Al	Mg
0.13	0.24	0.92	0.014	0.0035	0.0046	0.001	0.018	0.00132
0.13	0.28	0.87	0.0104	0.0021	0.0052	0.0016	0.018	0.0022

**Table 2 materials-11-02241-t002:** Summary of parameters of the grain growth equation.

*Q* (kJ/mole)	*K* _0_	*n*
31.4	590,064.8	0.26

**Table 3 materials-11-02241-t003:** Number of inclusions observed for four cases.

	Pinning Grain Boundary	Pinning Triple Point	Passing-Through Grain Boundary	Passing-Through Triple Point
MgO	3	1	1	0
MnS	0	0	3	1
MgO·Al_2_O_3_	1	0	0	0
MgO-MnS	1	2	1	0
MgO·Al_2_O_3_-MnS	1	1	1	0

**Table 4 materials-11-02241-t004:** Summary of grain mobility, boundary velocity, and ∆*F* of inclusions.

	MgO	MgO-MnS	MnS	MgO·Al_2_O_3_-MnS
*V* (μm/min)	0.5011	0.6974	2.2563	3.5371
Δ*F_n_* (kJ/m^3^)	9.8991	2.2298	5.4754	17.545
*m_p_* (10^−6^ m^4^·min/kJ)	0.0506	0.3128	0.4121	0.2016

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
