# Peer review of "Anisotropic Pinning-Effect of Inclusions in Mg-Based Low-Carbon Steel"

_materials, 2018, doi:10.3390/ma11112241_

Round 1
Reviewer 1 Report
!n abstract: correct it from 13 Mg to 13 ppm Mg.
The authors analysed the microstructure and explained their observations using available theory adequately.
Author Response
Comment 1: In abstract: correct it from 13 Mg to 13 ppm Mg.
Answer:
Thank you for your advice and we have revised the content according to your comment.
Original
| In this study, the effect of austenite grain size on acicular ferrite (AF) nucleation in low-carbon steel containing 13 Mg is determined. |
Revised: Page1/L1 | In this study, the effect of austenite grain size on acicular ferrite (AF) nucleation in low-carbon steel containing 13 ppm Mg is determined. |

Reviewer 2 Report
This paper discusses on “Anisotropic Pinning-Effect of Inclusions in Mg-based 2 Low-Carbon Steel”. The paper has been well organized and can be considered for publication by Journal of Materials; however, the following comments are recommended to be considered by authors before their work is published in the journal.
1) The authors considered the relationships between austenite grain size and the probability of AF nucleation as the purpose of this study. The main question is that what applications such relationships have in industry?
2) The authors concluded that the austenite grain boundary can strongly be pinned or passed by inclusions. The important question is what are the main reasons for pinning or passing a grain boundary by inclusions?3) In fig. 13, the authors showed in situ CSLM images of grain boundary transgranular by MnS inclusions. First, why isn’t it seen any inclusion inside holes? Second, EDS analysis is required to determine the type of inclusion.
4) The authors should clarify why MgO inclusions show the best pinning efficiency.
5) Fig. 13c is missing in the Figure 13.
Author Response
Response to Reviewer 2 Comments
Reference comments:
Comment 1: The authors considered the relationships between austenite grain size and the probability of AF nucleation as the purpose of this study. The main question is that what applications such relationships have in industry?
Response:
It can be applied to improve the heat affected zone during welding. Welding will cause the grain size in heat affected zone become coarse, and further reduce the strength and toughness of steel. In general, when the probability of AF nucleation increase, the toughness of steel will also increase because of its chaotic ordering microstructure. Small austenite grain size will also increase the strength of steel. In this study, we provided the relationship between the austenite grain size and the probability of AF nucleation. It can be controlled by optimizing the process parameter to get the high toughness or strength of steel according to different requirements.
Comment 2: The authors concluded that the austenite grain boundary can strongly be pinned or passed by inclusions. The important question is what are the main reasons for pinning or passing a grain boundary by inclusions?
Response:
Every inclusion type has different degree of pinning effect, and it will affect the pinning force. When the pinning force is larger than the driving force of austenite grain growth, the austenite grain boundary can strongly be pinned by inclusions. In contrary, if the pinning force is smaller than the driving force of austenite grain growth, the austenite grain boundary will pass through the inclusions and continues to grow. Thus, pinning or passing a grain boundary by inclusions would depend on the relative value of pinning force and driving force of austenite grain growth.
Comment 3: In fig. 13, the authors showed in situ CSLM images of grain boundary transgranular by MnS inclusions. First, why isn’t it seen any inclusion inside holes? Second, EDS analysis is required to determine the type of inclusion.
Response:
For the answer of first question, it is because that the depth of focus in CSLM is not enough to simultaneously focus on the grain boundary and inclusion. If we focused on the inclusion, then the area of grain boundary will be out of focus and not clear enough to see the grain boundary migration. In this study, we want to pay more attention on observing the grain boundary migration by in-situ images. So, we chose to focus on the grain boundary instead of inclusions. This is the reason why it can’t be seen any inclusion inside holes. Second, SEM-EDS and EBSD were utilized to determine the inclusion type, inclusion size and positioning by Vickers before heat treatment. The specimens were put in the chamber of high-temperature confocal scanning laser microscopy (HT-CSLM). Then, we found the positioning place first and start to do the heat treatment and calculate the velocity and driving force of each inclusion type during austenite grain boundary migration through HT-CSLM in situ observation.
Original | |
Revised: 3. Experiments Page4/L8 | Before heat treatment, SEM-EDS, EBSD, and ASPEX were utilized to determine the distribution of inclusion size, inclusion type, and positioning by Vickers. Then, specimens were heated at 1523 K for 600 s to calculate the velocity and driving force of each inclusion type during austenite grain boundary migration through HT-CSLM in situ observation. |
Comment 4: The author should clarify why MgO inclusions show the best pinning efficiency.
Response:
Thank you for your advice and we have clarified why MgO inclusions show the best pinning efficiency in the article.
Original | |
Revised: 3. Results and Discuss Page14/L13 | From Eqn. (4), the degree of pinning effect gets higher with smaller value of grain mobility. Among the Mg-based inclusions, MgO inclusions showed the lowest grain mobility; in other words, MgO inclusions effectively retard austenite grain growth while MnS inclusions do not. It means that the MgO inclusions have the best pinning effect than other inclusions. |
Comment 5: Fig. 13c is missing in the Figure 13.
Response:
Thank you for your advice and we have modified it to correct one.

Reviewer 3 Report
The article is interesting, however, several changes are required.
It should be clearly emphasis a novelty of the research. It is known that precipitates such as oxides significantly block the growth of grains in steels.
In the abstract, there is no ppm.
No information of sample preparation for austening grain size.
The quality of some figures is insufficient. For examples figures 1 and 3. How the austenite grain size was evaluated when grains borders are not visible in Fig. 1a, 1b.
Figures of the microstructure, please describe in the pictures the structure components listed in the text. Maybe it would be better to show individual components in SEM pictures. Were the calculations regarding the content of AF found experimental confirmation?
Why the considerations are carried out for different grades of steel.
There are no practical conclusions, recommendations regarding heat treatment of the analyzed steel.
Author Response
Comment 1: It should be clearly emphasis a novelty of the research. It is known that precipitates such as oxides significantly block the growth of grains in steels.
Response:
Thank you for your advice and we have revised the content according to your comment.
Original | |
Revised: abstract Page1/L11 | Next, the pinning ability of different inclusion types in low-carbon steel containing 22 Mg is determined. The in situ observation shows that it’s not every inclusion could inhibit austenite grain migration, the inclusion types has the different pinning ability. The grain mobility of each inclusion was calculated using in situ micrographs of CSLM to micro-analyze. Results show that the austenite grain boundary can strongly be pinned by Mg-based inclusions. MnS inclusions are the least effective in pinning austenite grain boundary migration. |
Revised: 3. Results and Discussion Page9/L22 | In general, inclusion could inhibit the austenite grain boundaries growth. In this study, we observed that it’s not every inclusion could inhibit austenite grain growth from in-situ observation of CSLM. This variation inspires us to investigate the relationship between the anisotropic pinning effect and inclusions. |
Comment 2: In the abstract, there is no ppm.
Response:
Thank you for your advice and we have modified to correct.
Original
| In this study, the effect of austenite grain size on acicular ferrite (AF) nucleation in low-carbon steel containing 13 Mg is determined. |
Revised: abstract Page1/L1 | In this study, the effect of austenite grain size on acicular ferrite (AF) nucleation in low-carbon steel containing 13 ppm Mg is determined. |
Comment 3: No information of sample preparation for austening grain size.
Response:
Thank you for your advice and we have revised the content according to your comment.
Original | |
Revised: 2. Experiments Page3/L21 | The sample preparation for austenite grain size is using sandpaper to gradually polish the steel specimens from #240, 600, 1200, 2000 and 3000. And then using diamond slurry to mirror polish them from 3 and 1 μm. To determine the effect of austenite grain size on AF nucleation, specimens of low-carbon steel containing 13 ppm Mg were used; these specimens were heated at 1473, 1573, or 1673 K for 30, 60, 180, or 300 s for austenitization. After heating treatment, the specimens were cooled to room temperature at a cooling rate of 100 K/s. The average austenite grain size was calculated by using OM Leica software based on ASTM E112. At least 300 grains for each condition were employed to measures average grain sizes and obtain reliable data |
Comment 4: The quality of some figures is insufficient. For examples figures 1 and 3. How the austenite grain size was evaluated when grains borders are not visible in Fig. 1a, 1b.
Response:
Thank you for your advice and the figures were changed to clear ones.
Comment 5: Figures of the microstructure, please describe in the pictures the structure components listed in the text. Maybe it would be better to show individual components in SEM pictures. Were the calculations regarding the content of AF found experimental confirmation?
Response:
Thank you for your advice. We have described the structure components in Figure 4 and Figure 9. In this study, the OM micrographs were observed different austenite grain size will affect the volume fractions of different phase. However, the volume fractions of different phase of each sample under different austenitization conditions were difficult to quantify from the OM micrographs. Thus, the probability of AF nucleation is used to quantify the ability of inclusions to induce AF at different austenitization holding times. The calculations regarding the content of AF is according to following equation:
Probability of AF nucleation = NAF/Ntotal x 100%
Where Ntotal and NAF are the total numbers of inclusions and inclusions with AF nucleation, respectively. After heat treatment, we used OM micrographs to find the microstructure of acicular ferrite around every inclusion. And separate the conclusions into two parts, one is inclusions without acicular ferrite formation, and the other is the inclusions with acicular ferrite formation. The number of inclusions is approximately 90–100 inclusions for statistical measurement of each sample for different austenitization holding times.
Original | |
Revised: 3. Results and Discussion Page6/L5 | The probability of pinning inclusions is defined by the following equation: Probability of AF nucleation = NAF/N total (1) Where Ntotal and NAF are the total numbers of inclusions and inclusions with AF nucleation, respectively. After heat treatment, we used OM micrographs to find the microstructure of acicular ferrite around every inclusion. And separate the conclusions into two parts, one is inclusions without acicular ferrite formation, and the other is the inclusions with acicular ferrite formation. We used approximately 90–100 inclusions for statistical measurement of each sample for different austenitization holding times. |
Comment 6: Why the considerations are carried out for different grades of steel.
Response:
According to Bhadeshia research, It shows that smaller austenite grain size is difficult to induce acicular ferrite nucleation. [1] The austenite grain size of low carbon steel containing 22 ppm Mg is smaller than low carbon steel containing 13 ppm Mg. The low carbon steel containing 22 ppm Mg is difficult to induce acicular ferrite than low carbon steel containing 13 ppm Mg. Thus, low carbon steel containing 22 ppm Mg is studied by pinning research, due to the high pinning effect. And low carbon steel containing 13 ppm Mg is studied by acicular ferrite research, due to the high probability of acicular ferrite nucleation. This is the reason why I chose different Mg contents of steel in this study.
Reference
[1] Bhadeshia H.K.D.H.; Christian J.W. bainite in steels. Metall. Trans. A. 1990, 21, 767.
Comment 7: There are no practical conclusions, recommendations regarding heat treatment of the analyzed steel.
Response:
Thank you for your advice and we have revised the content according to your comment.
Original | |
Revised: 4. conclusions Page14/L16 | The heat treatment condition at 1673 K for 60s has the highest probability of AF nucleation. |
